# Detection and Spatio-Temporal Distribution of Pinnatoxins in Shellfish from the Atlantic and Cantabrian Coasts of Spain

**DOI:** 10.3390/toxins11060340

**Published:** 2019-06-14

**Authors:** J. Pablo Lamas, Fabiola Arévalo, Ángeles Moroño, Jorge Correa, Susana Muñíz, Juan Blanco

**Affiliations:** 1Intecmar (Instituto Tecnolóxico para o Control do Medio Mariño de Galicia), Peirao de Vilaxoán s/n, Vilagarcía de Arousa, 36611 Pontevedra, Spain; farevalo@intecmar.gal (F.A.); amoronho@intecmar.gal (Á.M.); jcorrea@intecmar.gal (J.C.); susana.muniz.romero@xunta.gal (S.M.); 2Centro de Investigacións Mariñas (CIMA), Pedras de Corón s/n, 36620 Vilanova de Arousa, Spain

**Keywords:** pinnatoxins, shellfish, Spain, Atlantic, Cantabrian

## Abstract

For the first time, pinnatoxins have been detected in shellfish from the Atlantic and Cantabrian coasts of Spain. High sensitivity LC-MS/MS systems were used to monitor all the currently known pinnatoxins (A–H). Pinnatoxin G (PnTX G) was the most prevalent toxin of the group, but its metabolite PnTX A has also been found at much lower levels. No trend in PnTX G concentration was found in the area, but a hotspot in the Ría de Camariñas has been identified. The maximum concentrations found did not exceed 15 µg·kg^−1^, being, in most cases, below 3 µg·kg^−1^. The highest concentrations were found in wild (intertidal) populations of mussels which attained much higher levels than raft-cultured ones, suggesting that the toxin-producer organisms preferentially develop in shallow areas. Other bivalve species had, in general, lower concentrations. The incidence of PnTX G followed a seasonal pattern in which the maximum concentrations took place in winter months. PnTX G was found to be partially esterified but the esterification percentage was not high (lower than 30%).

## 1. Introduction

Pinnatoxins (PnTX) are cyclic imines (CI) that are structurally very related to spirolides and pteriatoxins. Together with other CI, these compounds are considered to be “fast acting toxins” because they produce, after intraperitoneal injection in mouse (and also by oral ingestion), the onset of neurological symptoms and their death in a very short period of time (a few minutes) [1,2]. PnTXs seem to have the highest oral toxicity among CI [3] probably due to their capacity to cross biological barriers and their stability during the digestive process [4,5]. However, to date, these toxins have not been linked unequivocally to human intoxications and, therefore, there is not a legal limit for them. An intoxication in China led to the discovery of PnTX A, but it was not clear if the toxin was the causative agent [6], as other agents that were present (e.g. *Vibrio*) could have produced the observed symptoms. No other human intoxication has been attributed to PnTXs to date, but recently, a bloom of the PnTX-producing species *Vulcanodinium rugosum* has been associated to skin lesions in Cuba [7].

Pinnatoxins were discovered in *Pinna attenuata* [8] in China. Some years after that, PnTX A to D were isolated from *Pinna muricata* from Japan and their structures elucidated [9,10,11]. In 2008, PnTX E, F and G were discovered in the Pacific oyster *Magellana gigas* from South Australia, and they were chemically and toxicologically characterised [1]. 

Pinnatoxins seem to be produced by only one dinoflagellate species, *Vulcanodinium rugosum* [6,12,13,14], which has been described from water samples of Mediterranean lagoons in France [15] and associated to PnTx in Japan, New Zealand, Australia and France [6,12,13,16].

Initially, the presence of PnTXs was detected in bivalves and, in some occasions, also in sediment and water samples from several areas of the Pacific Ocean, South China Sea [8], Japan [11,12,13,14,15,16,17], South Australia and New Zealand [1]. Subsequently, their presence was confirmed in New Zealand [18,19], and their distribution area was expanded to the Atlantic Ocean, where mainly PnTX G was found in Norway [19,20,21], Denmark [20], The Netherlands [20], Ireland [22], Canada [23] and in Portugal (at trace levels in bivalves) [20]. Pinnatoxins have also been detected in several places of the Mediterranean Sea, like France [6], Spain [24], Greece [25], Slovenia and Italy [20].

In most cases, PnTX concentrations found in the molluscs are very low (0.1–12 µg·kg^−1^ in the 96 samples analysed by Rambla-Alegre et al. [20], with maxima below 200, 120 and 110 µg·kg^−1^ in New Zealand, Norway and The Netherlands, respectively. Only in the Ingril lagoon, in the French Mediterranean, are these levels exceeded, reaching a maximum of 1244 µg·kg^−1^, and with base levels above 40 µg·kg^−1^ of PnTX G [6]. 

In Galicia (NW Spain), several studies conducted on bivalves, sediments and passive samplers have been unable to detect PnTXs, indicating that these toxins were either not present in the area or present at levels below the detection limits of the analytical equipment used. In most of the Cantabrian coast of Spain, as in Galicia, PnTXs have never been detected, to our knowledge.

In this study, we have examined 872 bivalve samples covering the Galician and Cantabrian coast of Spain (Figure 1) from April 2017 to April 2019 with a high sensitivity mass spectrometer, in order to know if PnTXs are present in the area and, in the case that they are present, their base level in bivalves.

## 2. Results

In the analysed samples PnTX G and PnTX A were unequivocally identified (Figure 2 and Figure 3). 

From April 2017 to April 2019, 872 samples were analysed and PnTX G was present in 261 samples (29.9%). The recorded levels were between 0.36 and 14.98 µg·kg^−1^. The other analysed PnTXs (Table 1) were not routinely monitored. 

In the case of mussels *M. galloprovincialis*, raft mussels seem to accumulate less PnTX G than wild ones. There were statistically significant differences in PnTX G concentration between species and between habitats (intertidal natural beds and raft cultured) (Figure 4, Appendix A). 

PnTX G was partially esterified, at least in all samples with concentrations above 2 µg·kg^−1^ of free toxin. The contribution of the esters to the total toxin content was, nearly always, below 22% (Figure 5). As the number of observations above 2 µg·kg^−1^ was low, only in a few cases could the percentage of esters be reliably computed. Mussels from natural beds contained a higher percentage of esters than those from culture rafts. It seems that there is a slight relationship between the percentage of esterification and the concentration of total PnTX G, but it was not statistically significant (p = 0.10, R^2^ = 0.14, Appendix A). 

Along the Atlantic and Cantabrian coast of Spain, the incidence of PnTX G was variable, with maximum levels in the Ría de Camariñas, followed by Corcubión and Ría do Barqueiro (Figure 6). The lowest levels were detected in the Ría de Ares. In the Basque Country, PnTX G was not detected. Therefore, no clear geographical trend was observed and the only clearly relevant aspect was the permanently high levels recorded in Camariñas, relative to those found in the whole area.

Even when the number of samples analysed from the spring-summer period was lower than that of the autumn-winter, it seems clear that PnTX G is especially prevalent in winter (Figure 7).

## 3. Discussion

This is the first report of PnTXs in the Atlantic and Cantabrian coasts of Spain. The presence of PnTX G and PnTX A has been demonstrated and it seems that no other known PnTX is present (Table 1). Pinnatoxin G had already been found in other areas of the Atlantic Ocean, like Canada [23], Norway [19], Denmark [20], the Netherlands [20], Ireland [22] and in one sample from the market in Portugal [20]. This study expands the distribution area of this compound and suggests that it could be present along all the Atlantic coast of Europe. The fact that it was not detected in the Basque Country and the lack of reports from the Atlantic coast of France prevents discarding the possibility of a gap of its distribution in that area. PnTX A is a product of the biotransformation of PnTX G in bivalves [1], and it has also been reported, in small amounts, in the same samples in which PnTX G was dominant, like those from Canada [23], Norway [19,21] and the Mediterranean area of France [6] and Spain [24]. In New Zealand, PnTX profiles dominated by PnTX F or by a combination of this toxin with PnTX G have been found, but it has been suggested that these toxins had been supplied by different dinoflagellate species [1]. 

The source of PnTXs in the Atlantic/Cantabrian coast of Spain has not been identified yet. It seems that a *Vulcanodinum* producing mostly PnTX G could be involved. Only one species of this genus has been described, but, in sight of both the genetic and toxin profiles, it has been suggested that *V. rugosum* could be a species complex [26]. Some strains of *V. rugosum* produce mainly PnTX G, like those from Mediterranean France (the area from which the species was described) [6], as well as a strain isolated from Japan [18], but some other strains do not produce PnTX G, for instance, most isolates from New Zealand and Australia produce mainly PnTX F [13] and most from China and Qatar produce PnTX H [27,28,29]. The toxin profiles obtained in this work would support the possibility of the Mediterranean strain as the source of toxins. Notwithstanding, the completely different seasonal pattern suggests that one or several different strains could be involved.

As in other areas, such as Canada [23], PnTX G was partially esterified. Notwithstanding, the proportion of esterified toxin was relatively low, being, in general, below 22%, in contrast with that observed in the mussel *Mytilus edulis* from Canada, where only a few observations were below 30% and with maximum values of 71%. Contrarily, in the mussel *M. galloprovincialis* and in the clam *Ruditapes decussatus*, from the Ingril lagoon in Mediterranean France, no esters were found, even when PnTX G attained high concentrations [6].

The levels of PnTX G during the studied period were low, in most cases, and close to the limit of detection of the equipment used. That is consistent with other studies in the Atlantic Ocean in which PnTX levels were below 150 µg·kg^−1^ in Norway [19,21], and below 41 µg·kg^−1^ in Canada [23]. The observed concentrations were about two orders of magnitude lower than those recorded in the Ingril lagoon (Mediterranean coast of France) [6], and also lower than in the Ebro Delta (Mediterranean Spain), where they attained 60 µg·kg^−1^ [24].

No clear spatial trend has been observed in the studied area. It appears that the PnTX prevalence diminishes towards the east, but the obtained data are too scarce to draw any solid conclusions. What seems clear, nevertheless, is the presence of a hot spot in the Ría de Camariñas in Galicia, where the observed concentrations were much higher than in the other areas. It seems also that the incidence has some relationship with the morphology of the sampling location, with relatively flat and low energy (due to river or tidal effects) areas attaining the highest concentrations.

A seasonal pattern, with maximum PnTX G levels during winter, was observed. This pattern has to be confirmed in future studies because the winter-spring months were over-represented in the current dataset (see Material and Methods). Nevertheless, in most months during the spring-autumn period, the detected PnTX G levels were low even when the number of PnTX G detections was relatively high, which suggest that the sampling bias did not affect the observed pattern. The information about the seasonal pattern of PnTXs is scarce, but some other areas seem to have a different seasonality. In the Ingril lagoon, for example, the maximum PnTX levels were attained during summer [6]. As commented above, when dealing with the origin of these toxins, it seems likely that a different species or strain of *Vulcanodinium* is involved, as it could be a species complex [26].

The studied species seem to have different capability to accumulate PnTX G. There are not enough observations to make reliable comparisons for most species, but among those in which 4 or more estimates were obtained, the wild mussel populations attained higher concentrations than both the cockle *C. edule* and the carpet shell clam *Vp. corrugata*, two species that share the intertidal habitat with the wild mussel.

PnTX G concentrations in raft-cultured mussels were substantially lower than in the wild mussel population. The most likely reason for this is that the concentrations of the pinnatoxin-producing species are higher in the intertidal area than in deeper waters. This is consistent with *Vulcanodinium rugosum*, the only known producer of PnTXs up to date, being, at least partially, an epibenthic/epiphytic species [30].

In conclusion, it seems that PnTX G is present in the Atlantic and Cantabrian coasts of Spain, but, apparently, in a patchy way. The risk of PnTXs in the area is low, as, in the broad range of samples analysed, the toxin levels were low and they were mainly restricted to wild mussels and to autumn-winter months. The observed differences between wild and raft-cultured mussels suggest that the producer species develop mainly in shallow waters. Esters are present but do not constitute an important proportion of the total PnTX G.

## 4. Material and Methods

### 4.1. Chemicals and Solvents

Acetonitrile (MeCN) was obtained from Merck (Darmstadt, Germany), methanol (HPLC grade quality) was obtained from Sigma-Aldrich Chemie GmbH (Steinheim, Germany) and ultrapure water was obtained from a Milli-Q A-10 system (Millipore Iberica, Madrid, Spain).

Ammonium hydroxide (NH_4_OH, 25%) and sodium hydroxide (NaOH > 99%) were obtained from Merck (Barcelona, Spain), and hydrochloric acid (HCL, 37%) from Panreac (Barcelona, Spain); all of them were analytical grade.

### 4.2. Sampling

Samples of several shellfish species were obtained from April 2017 to April 2019. Mussels *Mytilus galloprovincialis* (raft-cultured and wild) were used as sentinel organisms, being sampled, at least, weekly. Other bivalve species were only sampled when any kind of EU regulated toxin was detected in mussels. As a result of this sampling strategy and of the local abundance of the different species studied, mussel was the species most represented in the samples, followed by the cockle *Cerastoderma edule* and the carpet shell clam *Venerupis corrugata*. Other species have been analysed only on a few occasions (the razor clams *Ensis siliqua* and *Ensis arcuatus*, the clams *Polititapes rhomboides* and *Ruditapes philippinarum*, and the oysters *Magellana gigas* and *Ostrea edulis*, that were combined as “oysters”). As already commented, the sampled mussels proceed from two different habitats: raft-cultured and wild populations. Culture mussels are grown in ropes, typically 10-m long, hanging from rafts that are located in deep areas (deeper than the rope length), and wild populations grow in rocky substrates in the intertidal zone. Most other species also grow in the intertidal zone, with the exception of *P. rhomboids,* which is a subtidal species. 

Most samples were obtained from the Atlantic coast of Galicia (14 areas, from Baiona to Cedeira) but the Cantabrian coast was also sampled (7 areas, from Cariño to Basque Country) (Figure 1). The three sampling locations in Cantabria were jointly treated in data analysis, due to the low sample number obtained from each individual location, and the same procedure was followed for the Basque Country.

Until December 2018, not all samples obtained with the sampling scheme were analysed for PnTX, and most of the samples to be analysed were chosen because they contained 13-desmethyl SPX C. Therefore, even when this compound was frequently found, the analyses were neither completely at random nor systematic, and some bias in the results could exist. Starting in January 2019, PnTX G was monitored in all samples obtained.

Samples of mussel cultures from the Galician Atlantic coast are routinely taken with at least weekly frequency, but only a subset of those samples was analysed. The production areas from the north coast of Spain were sampled with a smaller frequency.

### 4.3. Extraction and Sample Preparation

For representative sampling, between 100 and 150 g of mussel soft tissues (previously rinsed with MilliQ water) were homogenised with a blade homogeniser. The extraction was carried out following the standardised operating procedure of the EU-RL for the determination of marine lipophylic biotoxins in molluscs [31]. Two grams aliquots of homogenised tissues were extracted by vortexing it twice with 9 mL of MeOH for 30 s. After each extraction, the slurry was centrifuged at 2000 g (4 °C) for 10 min. Both supernatants were combined adjusting the final volume to 20 mL with methanol. An aliquot of mollusc extract was filtered through 0.22 µm syringe filter (PVDF 0.22 µm Millipore, Madrid, Spain), then diluted with methanol (1:1 v:v) and finally analysed by LC-MS/MS system (injection volume, 5 µL). 

In order to determine the amount of PnTX G group toxins present in esterified forms, an aliquot of each extract was submitted to alkaline hydrolysis prior to the analysis. Hydrolysis was carried out following McCarron et al. [24] by adding 625 µL of aqueous NaOH 2.5 N to 5 mL of methanolic extract, homogenising in a vortex mixer for 30 s and heating the mixture at 76 °C for 40 min. After cooling down to room temperature, the extract was neutralised by adding 625 µL of HCl 2.5 N and homogenised in a vortex for 30 s. The resulting extracts were filtered by PVDF 0.22 µm syringe filter and analysed by LC–ESI-MS/ MS. The concentration of toxins in esterified forms was calculated from the difference between the concentration of free toxins in the extracts before and after the hydrolysis process.

Some samples were cleaned and concentrated by solid phase extraction using Strata-X 33 µm, 60 mg, 3 mL columns (Phenomenex, Torrance, CA, USA).

### 4.4. LC-MS/MS Analysis

The procedure used was validated and optimised in Intecmar for determination of lipophilic toxins following the Standardised Operation Procedure of the EU-RL for determination of marine lipophilic biotoxins in molluscs by LC-MS/MS version 5 (European Union Reference Laboratory for Marine Biotoxins et al., 2015), and was based on the method by Regueiro et al. [32]. The method is accredited following the norm UNE-EN ISO/IEC 1725 (Accreditation Nº 160/LE 394) for the EU regulated marine biotoxins.

A Waters (Barcelona, Spain) instrument consisting of a Xevo TQ-S triple quadrupole mass spectrometer equipped with ESI (electrospray ionisation) source operated in positive mode was used. For the chromatographic separation, an Acquity UPC BEH C18 (2.1 x 100 mm, 1.7 µm) (Waters, Barcelona, Spain) column was used at a flow rate of 400 µL min^−1^ and at 45 °C with a binary gradient of phase A (water) and B (MeCN 90%), both with 6.7mM NH_4_OH (pH 11). The gradient started at 25% B (for 1.66 min), followed by a linear increment to reach 95% B at 4.3 min and then held until minute 6.28. After that, in order to equilibrate the column, the following steps were added: return linearly to initial conditions in 2 min and held the initial conditions until minute 9. The injection volume was 5 uL. 

The mass spectrometer (MRM mode) with the ESI interface, was operated in positive ionisation mode with the following parameters: 1 V capillary voltage, 450 °C solvation temperature, 850 L h^−1^ N_2_ flow and 150 L h^−1^ (TQ-S) cone gas flow. Two transitions were selected to monitor each toxin (Table 1). 

PnTX G was identified and quantified by comparison with a reference certificate solution PnTX G (2.43 ± 0.11 µg·g^−1^ in methanol with 0.01% acetic acid) obtained from NRC (Canada). An individual stock solution of the toxin was prepared in methanol and stored in glass vials at −20 °C. Different working standard solutions were prepared by appropriate dilution in methanol and stored in glass vials for one week. 

Quality checks that were performed were focused on the linearity of the response and on the apparent recovery (true recovery plus matrix effect) in mussels, cockle and clams. The obtained linearity was good (R^2^ ≥ 0.99) and the apparent recoveries ranged from 77 to 80% in the three species tested. In each set of chromatographic runs, the same criteria for the EU regulated toxins was used (putting special attention to the variations in retention time, the ratio between quantifier and qualifier ions, the linearity of the calibration lines, the S/N ratio and the difference in slope between the calibration lines at the beginning and the end of each set). LOQ was 0.28 µg·kg^−1^ and LOD was 0.08 µg·kg^−1^.

In order to confirm the presence of PnTX G and PnTX A, some samples were cleaned and concentrated (x 10) with an SPE step (Phenomenex StrataX). The fragmentation spectra of the chromatographic peaks, tentatively identified as PnTX G and A in these cleaned/concentrated extracts, were obtained. The obtained sample fragmentation-spectra were compared with that corresponding to a reference toxin solution in the same conditions (PnTX G), and to the spectra reported in the literature (PnTX A). In all cases, a cone voltage of 60 V and collision energy of 36 eV was used.

### 4.5. Statistical Analysis

All the statistical analyses, ANOVA, Tukey HSD tests—for differences between species and habitats—and linear regression—for relationships between esterification and concentration—were carried out with R [33].

## Figures and Tables

**Figure 1 toxins-11-00340-f001:**
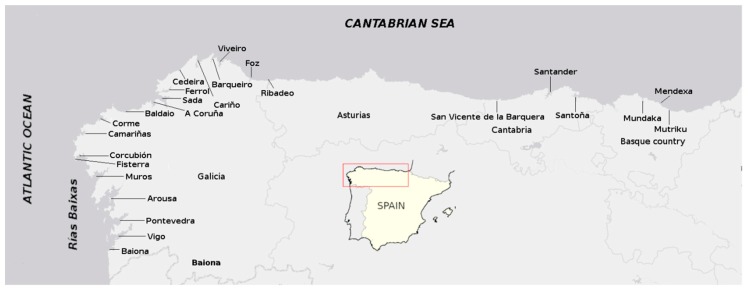
Sampling locations. Data from the three locations in Cantabria and the three in the Basque Country were grouped in the analysis. The inserted map shows the sampled area.

**Figure 2 toxins-11-00340-f002:**
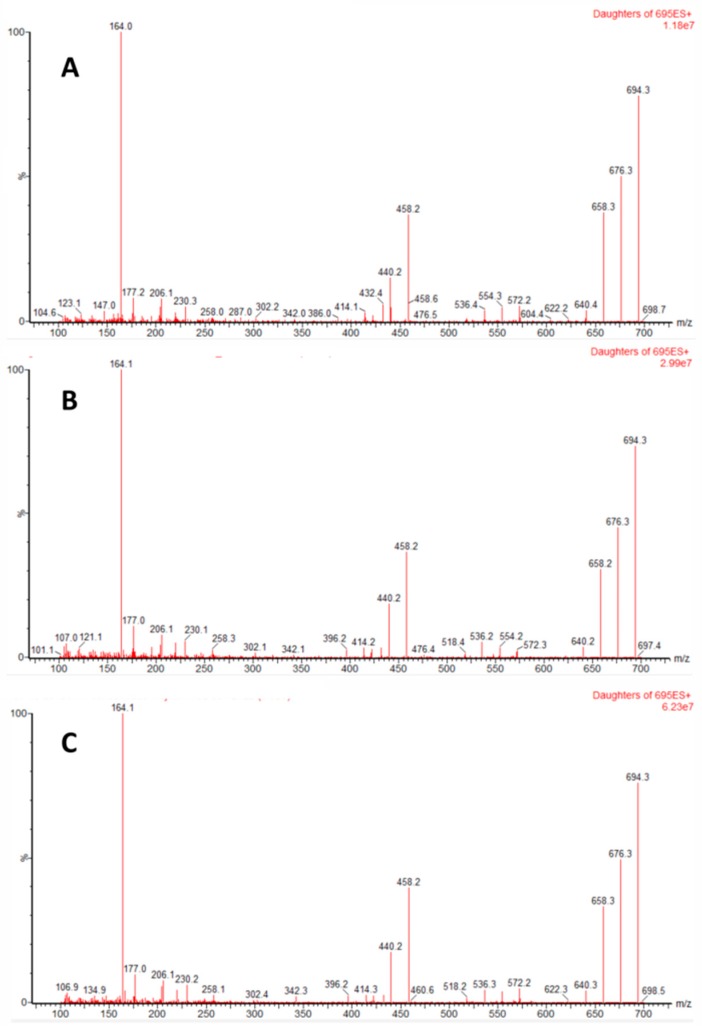
Fragmentation spectra for [M+H]^+^ ion of Pinnatoxin G (collision energy, 36V; cone voltage, 60V) for 1 ng mL^−1^ standard (**A**) and mussel extracts (**B**,**C**).

**Figure 3 toxins-11-00340-f003:**
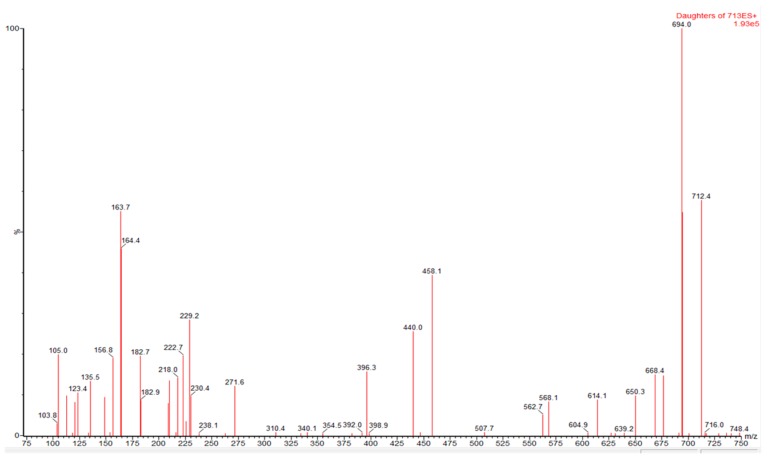
Fragmentation spectra for [M+H]^+^ ion of Pinnatoxin A (collision energy, 36V; cone voltage, 60V) in a sample of wild mussel from Camariñas.

**Figure 4 toxins-11-00340-f004:**
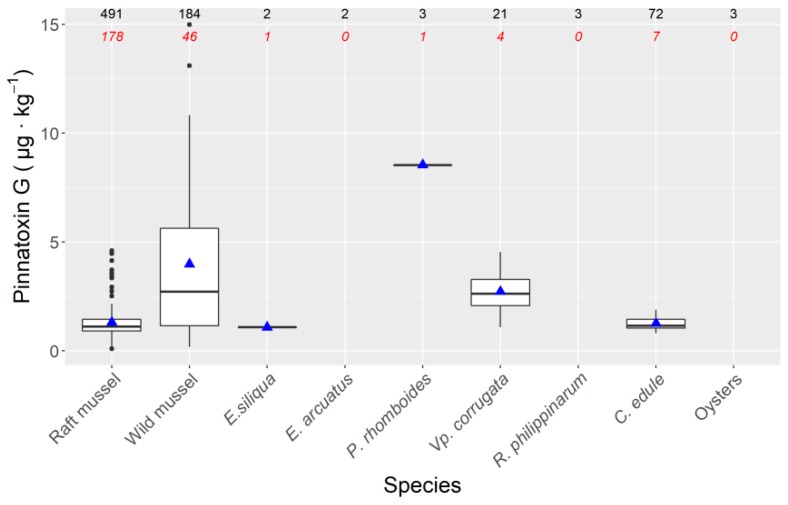
Pinnatoxin G concentrations recorded in the studied molluscs. Horizontal lines of the boxes are the 25%, 50% and 75% percentiles, and the triangle is the mean. The extremes of the vertical lines indicate the maximum and minimum values after excluding the outliers (data separated more than 1.5 IQR from the nearest quartile), that are represented as dots. The number of analysed samples are shown at the top of the figure. Those in italics are the number of samples in which Pinnatoxin G was detected.

**Figure 5 toxins-11-00340-f005:**
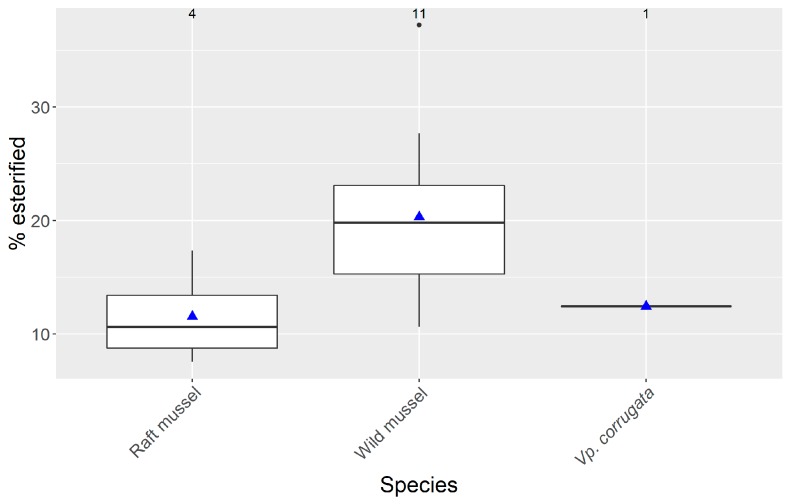
Percentage of esterified PnTX G in the studied molluscs in which concentrations above 2 µg·kg^−1^ of free PnTX were recorded. Graph details as in Figure 4.

**Figure 6 toxins-11-00340-f006:**
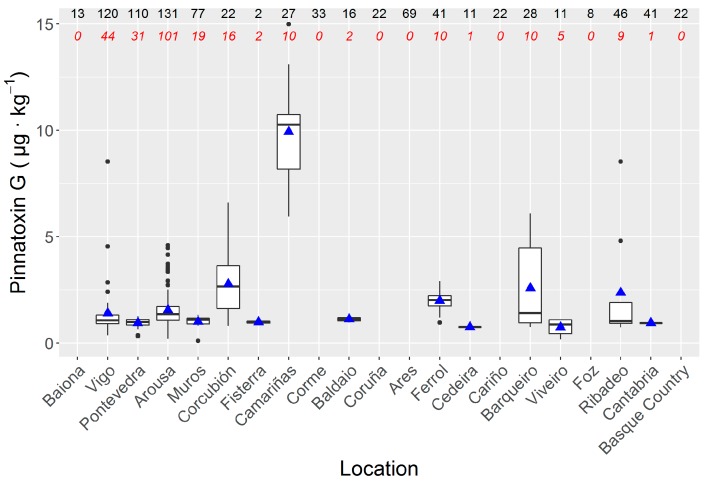
Pinnatoxin G concentration in the different areas sampled, ordered from south-west to east, covering from the south of Galicia to the Basque Country. Graph details as in Figure 4.

**Figure 7 toxins-11-00340-f007:**
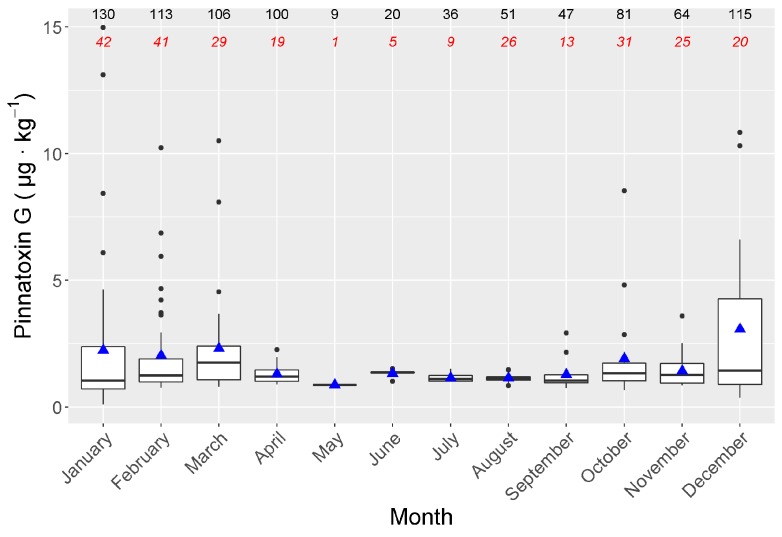
Seasonal variation of pinnatoxin G in the area. Graph details as in Figure 4.

**Table 1 toxins-11-00340-t001:** MRM transitions for the studied pinnatoxins.

Toxin	Ionisation	Parent Ion	MS/MS Transition		CE (eV)	Cone Voltage (V)
PnTX A	ESI^+^	[M + H]^+^	712.5>	164	50	60
			712.5>	440.3	36	60
			712.5>	538.3	36	60
			712.5>	694.5	36	60
PnTX BC	ESI^+^	[M + H]^+^	741.5>	164	50	60
			741.5>	458.3	36	60
			741.5>	723.5	36	60
PnTX D	ESI^+^	[M + H]^+^	782.5>	164	50	60
			782.5>	446.3	36	60
			782.5>	764.5	36	60
PnTX E	ESI^+^	[M + H]^+^	784.5>	164	57	60
			784.5>	446.3	41	60
			784.5>	766.5	57	60
PnTX F	ESI^+^	[M + H]^+^	766.5>	164	50	60
			766.5>	446.3	36	60
			766.5>	748.5	36	60
PnTX G	ESI^+^	[M + H]^+^	694.5>	164	50	60
			694.5>	440.3	36	60
			694.5>	676.5	36	60
PnTX H	ESI^+^	[M + H]^+^	708.5>	164.2	57	60
			708.5>	690.4	57	60

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
