# Peer review of "Detection and Spatio-Temporal Distribution of Pinnatoxins in Shellfish from the Atlantic and Cantabrian Coasts of Spain"

_toxins, 2019, doi:10.3390/toxins11060340_

Round 1

Reviewer 1 Report

Manuscript ID: Toxins-515259

Comments and Suggestions for Authors

The paper presents a study of pinnatoxins in shellfish and aims to assess the presence of these toxins in Galician and Cantabrian coasts of Spain. The theme of emergent toxins is of interest and pertinent to the scientific community. The literature survey is well documented and the quantification methodology is appropriate. However, the manuscript must be improved before it can be recommended for publication.

Specific comments:

1. English should be thoroughly revised, examples:

lines 22,  24, 173: please replace “they” changing accordingly the sentences;

line 9, please clarify “maximum concentrations were low…..”, is comparing with LOQs? Or with other quantification values from literature?

2. The manuscript title could be improved to be more attractive, a suggestion is to offer the study result in the title, example: Pinnatoxins were detected? With high values? Is a preliminary survey?

3. The key contribution could be improved including shellfish in the first sentence and clarifying the meaning of base levels low.

4. There is information missing in the abstract not giving an adequate picture of the entire paper. Please consider including a sentence with the names of pinnatoxins surveyed as well as the analytic methodology used.

5.  Introduction - line 28: please clarify “more likely” and the sentence “ Recently, ………to skin lesions in Cuba” could be moved to paragraph of dinoflagellate species (lines 35-37); line 38: it is not clear if the authors refer to pinnatoxins in shellfish or in other marine organisms, please clarify.

6. Results - line 65 - Please specify which pinnatoxins were not routinely monitored; line 66 - differences between species and between habitats for PnTX g concentrations were significant?; Figure 4: Please reformulate the legend of the figure to be more complete.

7. Discussion - line 92: please add a reference to support statement; line 118: compare results of this study with those obtained in [6].

8. Material and Methods:

- Line 150: compare  with literature the“low” levels obtained in this study;

- Section 4.2- this section describe the bivalve species sampled and the sampling areas, although the text is confuse needing reformulation;

- Section 4.3 - The toxin extraction and hydrolysis procedure used for pinnotoxins in this study is the same as for OA, DTXs, AZA, PTX and YTX. Please add information regarding the best conditions for hydrolysis of pinnatoxins esters, or explain the choice of those conditions.

- Section 4.4 - Please describe the quality control of LC-MS/MS analysis and include also detection limits and recovery results for each toxin; Did the authors tested matrix effects in the quantification of this compounds? Please add information about it.

- Section 4.5- In this section is missing which data were statistical analysed.

Author Response

Comments and Suggestions for Authors

The paper presents a study of pinnatoxins in shellfish and aims to assess the presence of these toxins in Galician and Cantabrian coasts of Spain. The theme of emergent toxins is of interest and pertinent to the scientific community. The literature survey is well documented and the quantification methodology is appropriate. However, the manuscript must be improved before it can be recommended for publication.

Specific comments:

1. English should be thoroughly revised, examples:

- lines 22,  24, 173: please replace “they” changing accordingly the sentences;

English has been revised

- line 9, please clarify “maximum concentrations were low…..”, is comparing with LOQs? Or with other quantification values from literature?

“Were low” was removed and only the absolute values are given.

2. The manuscript title could be improved to be more attractive, a suggestion is to offer the study result in the title, example: Pinnatoxins were detected? With high values? Is a preliminary survey?

Title replaced by a more precise one: Detection and spatio-temporal distribution of pinnatoxins in shellfish from the Atlantic and Cantabrian coasts of Spain

3. The key contribution could be improved including shellfish in the first sentence and clarifying the meaning of base levels low.

The Key contribution has been modified to fit the suggestion. It now reads:

This is the first report of Pinnatoxins in shellfish samples from the North coast of Spain. The dominance of pinnatoxin G and the association of this toxin mainly to shallow areas have been observed. Base levels of pinnatoxin G in the area are low in relation to those found in the Mediterranean Sea and New Zealand.

4. There is information missing in the abstract not giving an adequate picture of the entire paper. Please consider including a sentence with the names of pinnatoxins surveyed as well as the analytic methodology used.

The sentence “High sensitivity LC-MS/MS systems were used to monitor all the currently known Pinnatoxins (A-H)” has been added to the abstract

5.  Introduction - line 28: please clarify “more likely”

The sentence has been reworded as: “as other agents that were present  (e.g. Vibrio)  could have produced the observed symptoms.”

and the sentence “ Recently, ………to skin lesions in Cuba” could be moved to paragraph of dinoflagellate species (lines 35-37);

It seems that some introductory words were missing in the sentence. The aim of the sentence is to illustrate that PnTXs can actually be toxic, even when, only one toxic outbreak was (not unequivocally) associated to them. So, we have maintained the sentence in the same location but adding some introductory Word. It now reads:

“No other human intoxication has been attributed to PnTXs up to date, but recently, a bloom of the PnTX-producing species Vulcanodinium rugosum has been associated to skin lesions in Cuba [7].”

line 38: it is not clear if the authors refer to pinnatoxins in shellfish or in other marine organisms, please clarify.

The sentence has been modified by adding:

“in bivalves and, in some occasions also in  sediment and water samples, from…”

6. Results - line 65 - Please specify which pinnatoxins were not routinely monitored;

Two lines after, it is said that no other PnTx was routinely monitored. “detected” was replaced by “analysed” and a reference to the table in Material and methods has been added.

line 66 - differences between species and between habitats for PnTX g concentrations were significant?;

The sentence has been modified by adding “statistically significant”  before “differences”. We will try to add a table with the detailed results of the analysis as supporting information.

The table will be the R output of the analyses:

LMlogespecies<-lm(log(Pinna.G)~ESPECIE2,data=pinna)

> AOVlogespecies=aov(LMlogespecies)

> summary(AOVlogespecies)

             Df Sum Sq Mean Sq F value   Pr(>F)   

ESPECIE2      5  27.34   5.467   14.39 2.87e-12 ***

Residuals   231  87.77   0.380                    

---

Signif. codes:  0 ‘***’ 0.001 ‘**’ 0.01 ‘*’ 0.05 ‘.’ 0.1 ‘ ’ 1

544 observations deleted due to missingness

> TukeyHSD(AOVlogespecies)

  Tukey multiple comparisons of means

    95% family-wise confidence level

Fit: aov(formula = LMlogespecies)

$ESPECIE2

                                  diff        lwr         upr     p adj

Wild mussel-Raft mussel      0.7867092  0.4937221  1.07969617 0.0000000

E.siliqua-Raft mussel       -0.0776424 -1.8539975  1.69871274 0.9999956

P. rhomboides-Raft mussel    1.9849530  0.2085978  3.76130811 0.0186416

Vp. corrugata-Raft mussel    0.7231904 -0.1723991  1.61877983 0.1900166

C. edule-Raft mussel         0.0387578 -0.6438011  0.72131669 0.9999838

E.siliqua-Wild mussel       -0.8643516 -2.6548885  0.92618543 0.7346804

P. rhomboides-Wild mussel    1.1982438 -0.5922932  2.98878079 0.3906575

Vp. corrugata-Wild mussel   -0.0635188 -0.9869177  0.85988015 0.9999580

C. edule-Wild mussel        -0.7479514 -1.4666110 -0.02929167 0.0360024

P. rhomboides-E.siliqua      2.0625954 -0.4425232  4.56771390 0.1726591

Vp. corrugata-E.siliqua      0.8008328 -1.1796373  2.78130285 0.8543170

C. edule-E.siliqua           0.1164002 -1.7772914  2.01009181 0.9999760

Vp. corrugata-P. rhomboides -1.2617626 -3.2422327  0.71870748 0.4479777

C. edule-P. rhomboides      -1.9461952 -3.8398868 -0.05250356 0.0400554

C. edule-Vp. corrugata      -0.6844326 -1.7947077  0.42584256 0.4861949

Figure 4: Please reformulate the legend of the figure to be more complete.

The figure 4 has no legend. As no other figure has a legend we asume that the reviewer refers to the figure caption. The description of the wiskers and outliers has been added by means of the sentence:

The extremes of the vertical lines indicate the maximum and minimum values after excluding the outliers (data separated more than 1.5 IQR from the nearest quartile) , that are represented as dots.  

The corresponding references have been added

line 118: compare results of this study with those obtained in [6].

It is not clear for us, which we are requested to do.

The results from both studies are already compared throughout the manuscript for nearly every point in the discussion.

8. Material and Methods:

- Line 150: compare  with literature the“low” levels obtained in this study;

This subject was discussed in detail in lines 125-131. We prefer not to include that detailed information in this paragraph which is devoted to the general conclusions.

- Section 4.2- this section describe the bivalve species sampled and the sampling areas, although the text is confuse needing reformulation;

The section has been re-written

- Section 4.3 - The toxin extraction and hydrolysis procedure used for pinnotoxins in this study is the same as for OA, DTXs, AZA, PTX and YTX. Please add information regarding the best conditions for hydrolysis of pinnatoxins esters, or explain the choice of those conditions.

The extraction used is common for all lipophilic toxins. Using an extraction protocol optimized for pinnatoxins in the number of samples used in this study would be practically and economically impossible. In any case the recoveries are reasonable, being higher than 60% for the three bivalve species tested.

The hydrolysis was carried out following basically the procedure by McCarron et al 2012. The reference has been added. The procedure has not been optimized but, even if the optimization was performed, determining its efficiency would be impossible as no certified solution of esters is available.

- Section 4.4 - Please describe the quality control of LC-MS/MS analysis and include also detection limits and recovery results for each toxin; Did the authors tested matrix effects in the quantification of this compounds? Please add information about it.

The following text has been added to the material and Methods Section;

“Quality checks that were performed were focused on the linearity of the response and on the apparent recovery (true recovery plus matrix effect) in mussels, cockle and clams. The obtained linearity was good (R2  ≥ 0.99) and the apparent recoveries ranged from 77 to 80% in the three species tested. In each set of chromatographic runs, the same criteria than for the EU regulated toxins was used (putting special attention to the variations in retention time, the ratio between quantifier and qualifier ions, the linearity of the calibration lines, the S/N ratio, and the difference in slope between the calibration lines at the beginning and the end of each set). LOQ was 0.28 µg·kg-1  and LOD was 0.08 µg·kg-1.”

- Section 4.5- In this section is missing which data were statistical analysed.

The sentence has been modified. Now reads:

“All the statistical analyses, (ANOVA, Tukey HSD tests –differences between species and habitats–  and linear regression –relationship between esterification an concentration)– were carried out with R [35].”

Reviewer 2 Report

toxins-515259

The study entitled "Pinnatoxins in shellfish from the Atlantic and Cantabrian coasts of Spain"  shows, by a study using analytical chemistry techniques (LC-MS/MS), that pinnatoxins G and A - mainly PnTX G - is present in different molluscs on the North Atlantic coast of Spain, which had not been described so far. The authors unambiguously identify both toxins, and quantify the PnTX G content in different species, in its esterified form, with geographical disparities, and depending on the season.

This study is interesting because it provides information on the presence of a family of emerging toxins, which we realize is present in different geographical areas. The danger it represents for the consumer must raise the question of whether there should be European regulatory thresholds for PnTXs.

Minor corrections:

-          Assuming the number of times the words “pinnatoxin” and “pinnatoxins” appear in the article, I suggest to use the abbreviations PnTX and PnTXs.

-          L22-23: after intraperitoneal injection or through oral route in mouse

-          L23: death

-          L24-25: probably due to their capacity to cross biological barriers and their stability during digestive process

-          L45: In most cases, PnTX concentrations found

-          L53: s in Galicia, PnTXs had never been detected

-          Figure 1: place names are hard to read. Please increase their size. Insert a small map of Spain at top left of the figure to show that the region of concern is North of Spain and Galicia

-          Figures 2&3: adjust the size of the figures, particularly the width. Ameliorate the resolution. Increase the size of numbers, especially for the y-axis. Precise from where are the extracts in Fig.2B & 2C.

-           In Fig.2, a PnTx G was used and is shown (Fig.2A). Why isn’t there a standard for PnTx A in Fig.3 ?

-          µg kg-1 should be written µg.kg-1

-          Fig.4: latin names should be written in italics

-          L64-65: How can the minimum level of PnTx G (0.36 µg.kg-1) be lower than the LOQ (0.898 µg.kg-1) ? Please clearly indicate in the Mat & Meth section LOQ and LOD used in the study.

-          L70-76: what is the consequence of an esterified form of PnTx G ?

-          Fig.7: the seasonal variation of PnTX G content is in marked contradiction with what has been known so far, showing that PnTX presence in invertebrates peaks in August-September. How could you explain this discrepancy? Please insert this subject in discussion (following L136-137].

Author Response

The study entitled "Pinnatoxins in shellfish from the Atlantic and Cantabrian coasts of Spain"  shows, by a study using analytical chemistry techniques (LC-MS/MS), that pinnatoxins G and A - mainly PnTX G - is present in different molluscs on the North Atlantic coast of Spain, which had not been described so far. The authors unambiguously identify both toxins, and quantify the PnTX G content in different species, in its esterified form, with geographical disparities, and depending on the season.

This study is interesting because it provides information on the presence of a family of emerging toxins, which we realize is present in different geographical areas. The danger it represents for the consumer must raise the question of whether there should be European regulatory thresholds for PnTXs.

Minor corrections:

-          Assuming the number of times the words “pinnatoxin” and “pinnatoxins” appear in the article, I suggest to use the abbreviations PnTX and PnTXs.

“pinnatoxin” has been replaced by “PnTX” in all cases but when it appears at the beginning of a sentence and in special text, as abstract, key contribution, figure captions and conclusions.

-          L22-23: after intraperitoneal injection or through oral route in mouse

We have included “(and also by oral ingestion)” in the sentence, even when that sentence was focused on explaining why they were called “fast acting toxins”, which, as far as we know was because the effect on mice after intraperitoneal injection.

-          L23: death

Corrected

-          L24-25: probably due to their capacity to cross biological barriers and their stability during digestive process

Replaced

-          L45: In most cases, PnTX concentrations found

Replaced

-          L53: s in Galicia, PnTXs had never been detected

Replaced

-          Figure 1: place names are hard to read. Please increase their size. Insert a small map of Spain at top left of the figure to show that the region of concern is North of Spain and Galicia

Figure 1 has been modified as requested but the inserted map is not in the upper left because there was not enough space.

 -          Figures 2&3: adjust the size of the figures, particularly the width. Ameliorate the resolution. Increase the size of numbers, especially for the y-axis. Precise from where are the extracts in Fig.2B & 2C.

Unfortunately, the software we use does not allow to increase font size nor to export the spectra with a higher resolution. We prefer to maintain the original spectra than exporting the numeric data and replotting with a different software. We  think that the actual values in y-axis are not important to interpret the spectra, and, in fact, the only labels in the Y-axis are 0, % and 100. We also assume that, in the electronic version, the plots can be visualized in a larger size and all number could be more easily read..

 -           In Fig.2, a PnTx G was used and is shown (Fig.2A). Why isn’t there a standard for PnTx A in Fig.3 ?

Because, as far as we know, there is no commercial standard of PnTX A

 -          µg kg-1 should be written µg.kg-1

Replaced

-          Fig.4: latin names should be written in italics

Modified Figs. 4 and 5

-          L64-65: How can the minimum level of PnTx G (0.36 µg.kg-1) be lower than the LOQ (0.898 µg.kg-1) ? Please clearly indicate in the Mat & Meth section LOQ and LOD used in the study.

The text has been modified and the current LOQ and LOD are given in the Material and Methods section. The text should now read:

The LOQ in the previous version was established, for its use in the monitoring system, as the highest of the three LC-MS/MS (two Waters Xevo TQ-S and one Waters Xevo TQ-MS (less sensitive). In this study only one TQ-S system was used and now, the LOQ and LOD for that system (much lower than the values for the three systems) are given in the Material and Methods section.

-          L70-76: what is the consequence of an esterified form of PnTx G ?

Currently there is no practical consequence for the toxin monitoring system because no study of the toxicity of these derivatives has been carried out. Notwithstanding, it is clear that the molluscs in the area studied are able to produce these derivatives and consequently, that studying their toxicity could be interesting (or necessary if the PnTX levels increase). Additionally, the production of these compounds is interesting because they indicate a probable depuration route (as the one of okadaic acid and esteroids), and because, in that case, the kinetics of depuration could be substantially affected by the esterification rate. We don’t think that it would be positive to include these considerations in the manuscript as they would make it more complex and difficult to read.

-          Fig.7: the seasonal variation of PnTX G content is in marked contradiction with what has been known so far, showing that PnTX presence in invertebrates peaks in August-September. How could you explain this discrepancy? Please insert this subject in discussion (following L136-137].

The discrepancy had been already noted in the discussion (when dealing with the toxins source) and a possible cause had been pointed out. Notwithstanding, at the end of this paragraph the following text has been added :

”As commented above, when dealing with the origin of these toxins, it seems likely that a different species or strain of Vulcanodinium are involved, as it could be a species complex [27]”.

Round 2

Reviewer 1 Report

Dear authors,

the manuscript has been significantly 
improved and now warrants publication in Toxins.